# Breastfeeding during COVID-19: A Narrative Review of the Psychological Impact on Mothers

**DOI:** 10.3390/bs11030034

**Published:** 2021-03-14

**Authors:** Francisca Pacheco, Mónica Sobral, Raquel Guiomar, Alejandro de la Torre-Luque, Rafael A. Caparros-Gonzalez, Ana Ganho-Ávila

**Affiliations:** 1Center for Research in Neuropsychology and Cognitive Behavioral Intervention, Faculty of Psychology and Educational Sciences, University of Coimbra, 3000-115 Coimbra, Portugal; francisca.p.pacheco@gmail.com (F.P.); monicacsobral@gmail.com (M.S.); raquelguiomar18@gmail.com (R.G.); 2Department of Legal Medicine, Psychiatry and Pathology, Universidad Complutense de Madrid, Centre for Biomedical Research in Mental Health (CIBERSAM), 28040 Madrid, Spain; af.delatorre@ucm.es; 3Department of Nursing, Faculty of Health Sciences, University of Granada, 18071 Granada, Spain; rcg477@ugr.es; 4Mind, Brain, and Behavior Research Center (CIMCYC), University of Granada, 18011 Granada, Spain

**Keywords:** breastfeeding, COVID-19, mothers, maternal mental health, social support, postpartum, SARS-CoV-2

## Abstract

The COVID-19 pandemic has altered the normal course of life, with measures to reduce the virus spread impacting motherhood expectations and, in particular, breastfeeding practices. This study aimed to review evidence regarding the impact of COVID-19 on breastfeeding plans and how these relate to women’s psychological outcomes. Searches were conducted on PubMed and Web of Science for studies in English, Spanish, and Portuguese between January 2020 and January 2021. All study designs and pre-prints were considered. Twelve studies were included. Reports suggest that COVID-19 impacts differently on breastfeeding plans, which in turn leads to distinctive mental health outcomes. Positive breastfeeding experiences have been observed when mothers perceive that they have more time for motherhood, which may be associated with better mental health outcomes. Negative breastfeeding experiences have been observed when mothers are separated from their newborns, when mothers struggle with breastfeeding, or when mothers perceive decreased family and professional support, which seems to be associated with worse mental health outcomes. These preliminary results highlight the need for further research into the association between COVID-19, breastfeeding expectations, and maternal mental health. Filling this gap will foster the development of guidelines and interventions to better support mothers experiencing the obstacles of COVID-19 pandemic.

## 1. Introduction

The COVID-19 outbreak, caused by the severe acute respiratory syndrome coronavirus 2 (SARS-CoV-2), was identified at the end of 2019. The pandemic has been affecting every aspect of life around the world since then [1]. As the virus transmission occurs through respiratory droplets and mainly in close contact, amongst the proposed global health measures to reduce its spread are the implementation of lockdowns, confinement, and social distancing [2]. Such measures have resulted in the separation of mothers and infants after birth, particularly in cases of suspected or confirmed COVID-19 positive mothers, preventing mother-baby close contact including breastfeeding [3]. COVID-19 related policies (e.g., separation of the mother-infant) are expected to negatively impact maternal mental health outcomes, representing a critical challenge for the development of recommendations within the maternal health care services [4].

Breastfeeding has been the most globally recommended method for infant feeding [5]. Nonetheless, breastfeeding practices differ across cultures, with exclusive and continued breastfeeding presenting higher rates in low and middle-income countries and formula being most common in Western Europe, Australia, and North America [6]. Besides culture and sociodemographic conditions, the decision regarding breastfeeding is also influenced by other psychosocial and policy factors, enhancing decision making in favor (e.g., increased maternity leave, perception of support from the partner in newborn care tasks) or against breastfeeding (e.g., precarious work). Additionally, breastfeeding is frequently a challenging experience for mothers due to lactation difficulties or discomfort (e.g., sore nipples) [7]. The availability of support, in particular professionalized care, is frequently critical for success [8,9]. A qualitative study conducted in Sweden illustrates well how breastfeeding decisions in high-income countries emerge as the result between social expectations and the sense of self-efficacy as a parent [10]. Qualitive content analysis from online forums and webpages showed the emergence of three defining themes that contribute to the decision of breastfeeding or adopting formula milk. The defining themes were “striving to be a good mother”, “striving for your own well-being”, and “striving to discover your own path”. In this sense mothers balanced between the need to be a good mother and feeling well while attempting to find their own path.

During global infection outbreaks such as the COVID-19 pandemic, however, this sense of self-efficacy and confidence in decision making is frequently put-on hold. For example, COVID-19-positive mothers may face clinical impediment for breastfeeding; there is limited professional support during the first days after birth to help parents deal with negative experiences (e.g., difficult latching, sore nipples, thrust infections, tongue ties), and social distancing measures impair the familial or social environment that support the young parents’ journey [11]. On the contrary, there may be an increase in breastfeeding practices as mothers previously not planning to breastfeed change their plans because the pandemic extends their maternal leave or increases their presence at home [12], or breastfeeding is even seen as a protective measure to boost the newborns’ health. Additionally, extensive media coverage can induce fears in new parents and clinicians related to both formula and breastfeeding practices, changing perspectives and assumptions during a global health crisis. For instance, parents who read articles on social media regarding feeding methods safety might switch their decision regarding breastfeeding or not, even when the guidelines recommend otherwise [13].

The peripartum represents a particular period of vulnerability, characterized by an increased maternal sensitivity added to intense, novel, and overwhelming caregiving tasks [2]. The extensive physiological changes that occur during the peripartum period include unprecedented hormonal fluctuations that increase the vulnerability of women to neuropsychiatric disorders [14,15,16]. In particular, during the first months after delivery women are 22 times more likely to develop a mental health disorder than in any other period of their lifetime [17]. Mental health disorders in the peripartum are also associated with poverty, physical health, the quality of the intimate partner relationship, violence, extended family support, and other forms of sociodemographic disadvantages [18,19]: circumstances that are broadly inflated across countries in times of global pandemics such as the COVID-19.

Adding to the typical challenges of the perinatal period, the COVID-19 pandemic has been interfering with peripartum women’s emotional well-being. A recent rapid survey conducted in April–May 2020, Canada, has identified a significant increase in mental health symptomatology, particularly in depressive and anxiety outcomes. This survey assessed 900 women, 520 who were pregnant and 380 in the first year of the postpartum period by the time of the study. Results showed that 15% (pre-pandemic) and 40.7% (current) women significantly indicated depression. Additionally, 29% (pre-pandemic) and 72% (current) women presented moderate to high anxiety [20].

Besides the biological, sociodemographic, and epidemiological variables that contribute to women’s mental health during the perinatal period, breastfeeding experiences are known to impact postpartum mental health, positively influencing self-reported and physiological measures of mood, stress, and maternal care [21,22]. On the contrary, un-met prenatal expectations regarding breastfeeding seem to contribute to the correlation between lack of breastfeeding and postpartum depression for middle and high-income women [23], suggesting that beyond following breastfeeding recommendations, supporting women’s idiosyncratic expectations regarding pregnancy, birth, and postpartum plans may improve maternal mental health.

The COVID-19 pandemic has also intensified the concerns of mothers regarding breastfeeding [24]. The fear of vertical transmission through breastfeeding is currently a core concern of new mothers, and the evidence is still conflicting [25]. Whereas a recent study that included a sample of six women infected with COVID-19 showed no evidence of the virus in breast milk samples [26], two other studies [27,28] found SARS-CoV-2 in women’s breast milk. Despite the mixed results, a recent review of 26 global and governmental guidelines reports that contrary to the initial measures adopted early in 2020, the overall current consensus recommends breastfeeding or expressed milk, including when mothers are infected [29]. In the specific case of mothers who tested positive for COVID-19, three possibilities should be considered: the use of formula or donor milk when the mother or the infant is too ill, the use of expressed breast milk, or the adoption of breastfeeding with precautions (e.g., use of surgical mask [30]). Despite the risks, the Word Health Organization (WHO) supports the continuation of breastfeeding as well as postpartum skin-to-skin contact as long as the necessary precautions are adopted [31,32,33]. These general guidelines should then inform shared decisions to be taken by parents and discussed with their perinatal health providers [31], outweighing the risk of transmission and the advantages of breastfeeding [26].

Gathering data about the impact of the pandemic on breastfeeding practices, women’s expectations, their perception of social support, and on their psychological well-being is essential to develop guidelines aimed to inform maternal health care practices and help mothers to overcome the challenges inherent to the perinatal period during global stressful events. While there is a growing concern about the impact of COVID-19 social distancing/confinement policies [34], its impact on maternal mental health outcomes (e.g., self-efficacy, sadness, frustration) as a consequence of the altered breastfeeding expectations has not yet been adequately mapped. Thus, the aim of this study is to review how the COVID-19 pandemic has been impacting breastfeeding practices/plans and mothers’ expectations and to investigate the effect that un-met breastfeeding expectations have on women’s mental health.

## 2. Methods

The current narrative review resulted from a search for publications related to COVID-19 psychological impact on breastfeeding. Searches were undertaken in October 2020 and updated in January 2021 on PubMed/Medline and Web of Science. Additionally, due to the fact that the COVID-19 pandemic started recently and that there are numerous published but also still unpublished studies, we searched for pre-prints in Medrxiv. The following Medical Subject Headings (MeSH) search terms were used: (Covid-19 OR SARS-CoV-2 OR coronavirus) AND (breastfeeding OR “breast milk”) AND (“mental health” OR anxiety OR depression OR impact OR “perinatal mental health” OR stress). Studies were considered if they had been conducted between January 2020 and January 2021 and were written in English, Spanish, or Portuguese. Our inclusion criteria were studies that enrolled women during the postpartum period and accessed breastfeeding practices during the COVID-19 pandemic. We included empirical studies of primary research, observational or participatory studies, qualitative studies, and reviews and meta-analyses. Opinion pieces, editorials, conference abstracts, study or review protocols, and book chapters were not selected. Two reviewers (FP and MS) performed the searches and the screening for articles. The first screening was conducted independently, which was not the case for the full-text phase. Both extracted the data. In order to review manuscripts in depth, we contacted the authors for missing data (additional sample results from the group of women who stopped breastfeeding, detailed descriptions of other free-text responses of reasons behind breastfeeding cessation) every time it would be of benefit for the current work. After selection and assessment of the articles, we performed the narrative synthesis focused on the overall breastfeeding experiences related to COVID-19.

An overview of the selection process, according to the Preferred Reporting Items for Systematic Reviews and Meta-Analyses (PRISMA) flow diagram [35], is presented in Figure 1.

## 3. Results

A total of 104 peer-reviewed studies and 68 pre-prints were retrieved. During the first screening (title and abstract review), 30 studies in total were included, and after the full-text screening 11 remained (10 published reports and one pre-print). For the relevant studies, we also hand-searched the reference list of the selected articles (descendant search strategy) and included one additional article. In total, 12 studies were selected for the review. The main characteristics and findings of reviewed studies are presented in Table 1.

Considering the available literature and the goal of the present review, we organized the impact of COVID-19 on breastfeeding according to the following categories: (i) breastfeeding practices, (ii) safety concerns related to SARS-CoV-2 transmission, (iii) perceptions of support, and (iv) psychological outcomes of mothers.

### 3.1. Impact of COVID-19 Pandemic on Planned Breastfeeding Practices

#### 3.1.1. General Impact of COVID-19 on Planned Breastfeeding

Overall, no significant changes regarding breastfeeding practices were reported, except in frequency or duration. An online survey conducted in primary care services in Belgium collected data from 6470 women, of which 3823 were breastfeeding [36]. This study found that 91% of the women surveyed stated that they had not changed the feeding plan due to the coronavirus pandemic. In fact, 97% of women already breastfeeding stated to have not even considered ceasing breastfeeding. Contrarily, 82% reported that they had increased the breastfeeding frequency due to being at home for extended periods (a consequence of the lockdown) and because they believed that breast milk would protect their infants against COVID-19. Regarding breastfeeding duration, 55% of mothers mentioned to have considered extending the breastfeeding period as a result of the pandemic. On the other hand, 18% of women had reduced the breastfeeding frequency due to a reduction in the quantity of breast milk which they associated with worries related to the global pandemic and to the increased concerns about childcare responsibilities at home. When the mothers had previously experienced breastfeeding, 86% stated that COVID-19 did not impact their decision as well [36].

According to this survey [36], 88% (*n* = 112) of the women that stopped breastfeeding in the four weeks prior to the survey stated that this cessation was not caused by COVID-19. In contrast, 12% (*n* = 15) of women attributed breastfeeding cessation to COVID-19, specifically due to the consequences of the lockdown, such as the increased caregiving responsibilities and the need to work from home (*n* = 6), the increased workload and the inability to combine work with breastfeeding (*n* = 4), and the reduced milk production due to COVID-19 concerns (*n* = 4). Additionally, the survey offered free-text responses, and there women attributed breastfeeding cessation to difficulties in managing family care (*n* = 4), no milk production caused by stress and time pressure resulting from being a healthcare provider (*n* = 1), psychological burden (*n* = 1), and lack of guidance/information (*n* = 1) resulting from the COVID-19 pandemic.

An additional online survey conducted in the United Kingdom by Vasquez-Vasquez et al. [37] collected data from 1365 women in the postpartum period (infants ≤ 12 months of age). Forty percent of the mothers who were breastfeeding reported a change in feeding frequency: 30% (*n* = 234) increased feeding frequency and 10% (*n* = 73) decreased feeding frequency. Additionally, regarding the duration of breastfeeding practices, 68% (*n* = 524) reported no changes in the duration of each feed, whilst 17% (*n* = 120) reported an increase and 15% (*n* = 117) a decrease. Finally, 4% (*n* = 42) of the surveyed women reported to have stopped breastfeeding, but no further detailed information was offered regarding the reasons behind this. According to the same study, 13% of the mothers changed their plans concerning the infant’s feeding due to the lockdown, and this was true both for mothers that gave birth before (*n* = 138) and during the lockdown (*n* = 39). The most common reason for changing the breastfeeding plan was the lack of support (*n* = 21), especially in-person consultations, leading these mothers to either express milk, introduce formula, or stop breastfeeding. Some women had breastfeeding problems (*n* = 6), namely, infants having physical conditions such as tongue-tie. Although these problems were not directly related to COVID-19, their treatment was not conducted due to the pandemic [37]. The online survey by Brown and Shenker [13] conducted in the United Kingdom collected data from 1219 breastfeeding mothers (infants ≤ 12 months of age) and reported that although 18.9% (*n* = 230) of the women surveyed stated to have stopped breastfeeding, this was due to other reasons unrelated to COVID-19. Although the majority of the sample was constituted by women who gave birth during the pandemic (59.4%; *n* = 724), the main reason presented for interrupting breastfeeding was the overall lack of professional support, regardless the time of birth (pre vs. during the pandemic). That is, giving birth before or during the pandemic did not influence the decision to stop breastfeeding. When observing mothers that reported that COVID-19 had influenced their decision towards breastfeeding cessation, the lack of face-to-face professional support (70.3%) was the main reason, followed by worries about the safety of breastfeeding (20.9%) and the presence of symptoms of COVID-19 (6.5%).

An app-based survey with mixed format questions, conducted in the United States with 258 pregnant women [38], found that only one woman (1,1%) changed her initial plan to breastfeed moving to formula, and, according to the mother, the decision was influenced by the presence of depressive symptoms and not specifically due to the COVID-19 pandemic.

An Italian case-control study [44] focused on breastfeeding initiation practices, gathering data on 152 breastfeeding mothers during the pandemic (study group) and 147 breastfeeding mothers who delivered in 2019 (control group). The authors found that exclusively breastfeeding initiation rates were significantly lower in the study group, who adopted more complementary feeding practices.

#### 3.1.2. Impact of COVID-19 on Planned Feeding Methods—Moving from Other Methods to Breastfeeding

According to the app-based survey conducted by Burgess et al. [38], only five women (2.6%) reported changing their infant’s feeding plan due to COVID-19 from formula to breastfeeding. The reasons presented for this change were formula shortages (*n* = 1), cost of formula (*n* = 1), fear regarding the possibility of formula contamination (*n* = 1), and believing that breast milk was a better option for protecting their infant (*n* = 4). No other studies emerging from our search reported on the change from formula to breastfeeding, either due to COVID-19 or other unrelated factors.

#### 3.1.3. Impact of COVID-19 on Mothers Who Tested Positive for COVID-19

Two studies exclusively evaluated the feeding plan of mothers infected with SARS-CoV-2 [39,43]. Firstly, a multicenter cohort study was conducted with 125 COVID-19-positive mothers in neonatal intensive care units in Turkey, whose data were collected by the neonatologist using the electronic case report form (eCFR). The authors gathered information on peripartum risk factors, demographic, epidemiologic and clinical features, treatment strategies, and breastfeeding history. Results showed that the most common feeding practice adopted by COVID-19-positive mothers was formula (*n* = 71, 56.8%), followed by expressed breast milk (*n* = 45, 36%). Nonetheless, some mothers still opted for exclusive breastfeeding with caution (*n* = 9, 7.2%). The study authors argued that the following reasons might have affected the rate of breastfeeding: isolation, the anxiety of parents and clinicians about the possibility of breast milk contamination and the health status of mothers [39].

Secondly, an observational longitudinal cohort study in the United States evaluated 85 mother–newborn dyads from three hospitals [40]. The authors conducted a telephone survey and gathered data on feeding plan (before delivery, during hospitalization, and at home). Four options of the current feeding method were presented (breastfeeding, expressed breast milk, formula, or mixed feeding) across measurement occasions. When changes in feeding plan were identified, mothers were asked about the reason and the association between this and COVID-19. Separation due to COVID-19 and difficulty latching were indicated as the reasons for feeding plan changes in 30 mothers (35.3%; 24 mothers who were separated from the infant, and six who were not) from predelivery to hospital or home. Twenty-three mothers (27.1%; six separated, 17 not separated) did not report changes in feeding from predelivery to hospital or home. For mothers who were separated from the infant, breastfeeding was the most common for the predelivery plan (*n* = 28; 57.1%), formula feeding during hospital feeding (*n* = 40; 81.6%), and mixed feeding during home feeding (*n* = 25; 51.0%). For mothers-infants who were not separated, breastfeeding was the most common for the predelivery plan (*n* = 23; 63.9%), mixed feeding during hospital feeding (*n* = 16; 44.4%), and mixed feeding during home feeding (*n* = 25; 58.3%). COVID-19 had an important role in changes in the feeding plan from predelivery to hospital and/or home feeding. Nonetheless, changes to the plan due to COVID-19 differed significantly depending on the separation status (49.0% separated vs. 16.7% not separated).

#### 3.1.4. Impact of COVID-19 on Breastfeeding Experience

The online survey by Brown and Shenker [13], showed that breastfeeding experiences were influenced by the COVID-19 pandemic, either positively (41.8%; e.g., spending more time at home, experiencing less social pressure, and fewer visitors), negatively (27%; e.g., less perceived support, worries about safety of breastfeeding, and isolation), or neutrally (29.5%). Women who perceived the breastfeeding experience during the pandemic as more challenging were less likely to describe themselves as ready to stop breastfeeding. Furthermore, 1.7% of women were not able to precise the perceived impact. These authors also found a strong significant association between breastfeeding perceptions and current feeding method (breastfeeding vs. not breastfeeding). While 48.8% of those still breastfeeding felt the experience to be positive, just 15.9% of those no longer breastfeeding felt the same way. When the time of birth was before the pandemic, 50.1% of women rated their experience of breastfeeding as positive, and 10.7% as negative. When giving birth after the pandemic, only 36.2% of women rated their breastfeeding experience as positive, while 34.8% rated it as negative [13].

According to this study, not just the pandemic but also the consequent lockdown significantly impacted breastfeeding experiences. In fact, the majority of mothers who were breastfeeding perceived that COVID-19 impacted negatively due to the need to stay at home (50.8%), not receiving visits from their relatives (52.1%), the need to cancel attendance to support groups (72.8%), not being able to go to health care facilities (70.6%), and the excessive time to focus on breastfeeding. However, a considerable number of women perceived a positive impact of COVID-19 because they had the opportunity to have their older children at home (30.4%), they had fewer visits in the hospital (25%), and they perceived that they had more time to exclusively focus on breastfeeding. In comparison to women who were not breastfeeding, breastfeeding women perceived the overall impact as more positive [13].

An online Australian survey conducted by Hull et al. [40], consisting of nine open-ended questions, assessed mental health outcomes in 340 participants, of which 336 were mothers. The authors suggested that the pandemic influenced how other problems were presented and dealt with. In particular, the study shows 292 reported concerns that are associated with breastfeeding. Of these, the most common were “insufficient milk or inadequate weight gain”, “painful breasts or nipples”, “relactation”, and “infant formula supplementation”. Although COVID-19 was not always the predictor of these difficulties, the authors stated that in many cases COVID-19 was responsible for the concerns and their expression.

The United States study by Snyder and Worlton [42] conducted semi-structured interviews and collected data from 29 currently breastfeeding mothers, mainly mothers of multiple children (*n* = 19). Most mothers perceived a positive impact of the pandemic on breastfeeding as it increased the duration of the maternity leave and facilitated its practice. In addition, some of the mothers preferred to continue breastfeeding since it was a safer option (e.g., fear of formula shortages). Additionally, these mothers expressed that if this pandemic had occurred during their first breastfeeding experience, the impact would have been perceived as more negative.

A United States case series study interviewed and analyzed three mothers’ experiences regarding birth and breastfeeding [45]. Two mothers were combining direct breastfeeding and expressed milk (pumping), and one mother was exclusively breastfeeding. All mothers reported disruptions to the breastfeeding experience as a consequence of the COVID-19 pandemic. Specifically, mothers reported changes to the lactation support (telehealth). However, these mothers also pointed out some positive aspects, namely, having to be at home as it had benefited breastfeeding.

### 3.2. Breastfeeding Safety Concerns

According to the study by Hull et al. [40], the main safety concern regarding breastfeeding, as reported by mothers that tested positive for SARS-CoV-2 (*n* = 25), was the fear of transmission of the virus and of infecting the baby or causing harm to them, which increased mothers’ anxiety. In the study by Brown and Shenker [13], although 13.2% of the surveyed women reported to worry about the safety of breastfeeding during the pandemic, the majority reported not being currently worried (80.3%). Importantly, information about the risk of breastfeeding was found to influence the current infant’s feeding practice by mothers. Namely, women who had stopped breastfeeding were more likely to have received information regarding breastfeeding risks from health professionals, friends, or family [13].

### 3.3. Impact of COVID-19 on Perceptions of Support

Regarding support, Brown and Shenker [13] found that 39.8% of women perceived to have received enough practical support, and 36% perceived to have received enough emotional support from health professionals. Breastfeeding mothers were more likely to report having received this support. Importantly, mothers who gave birth during the pandemic felt less supported when compared to those who gave birth before the pandemic [13].

Additionally, according to the findings by Ceulemans et al. [36], 39% of breastfeeding women reported that the COVID-19 pandemic impacted the social support received during the breastfeeding period. Of these, 87% felt less supported by family/friends, 86% felt less supported by perinatal organizations, and 68% felt less supported by maternity assistance at home. Moreover, this impact was more likely to be reported in women who were breastfeeding for the first time [2].

In Worlton’s study [42], the COVID-19 pandemic changed the way in which support was obtained for women who were breastfeeding. Across mothers, the main source of support observed was family members. Regarding informational support, one third of mothers perceived to have not received enough effective support for latching problems. Due to the lack of informational support, mothers of multiple children benefited from having previous breastfeeding knowledge. In the same sample, feelings of frustration and isolation were reported due to the decrease in in-person support networks.

### 3.4. The Psychological Outcomes of Mothers

COVID-19 and the decreed lockdowns have affected new mother’s mental health. Results of the survey conducted by Hull et al. [41] showed that mothers were experiencing isolation (*n* = 31), anxiety or stress (*n* = 53), and needing reassurance (*n* = 63). As observed in the qualitative reports, mothers’ preoccupations about breastfeeding caused significant anxiety and were related to the belief that breastfeeding could protect their infants against the virus.

The online survey conducted by Vasquez-Vasquez et al. [37] reported that for most participants the expectations of motherhood were not met, leading to the feeling of having lost experiences that cannot be regained.

One case report emerged [41], describing a COVID-19 positive 26-year-old Indian woman in the postpartum period, who gave birth prematurely (at the 28-gestational-week), and whose newborn was in the neonatal intensive care unit (ICU). The mother reported loneliness, distrust regarding the COVID-19 results, belief of unjust isolation, loneliness and lack of family support, fear, anxiety, anger, stress, and depression. Additionally, the young mother reported willingness to breastfeed (“I want to breastfeed my baby”) associated with the perception of lost opportunity of feeding the infant (“I have not breastfed him. I just want to go back, see him and breastfeed him”), causing feelings of guilt and decreasing the perception of efficacy as a mother.

Additionally, in study [42] mothers reported experiencing heightened stress during breastfeeding, not only due to the lack of support caused by COVID-19 pandemic but also related to concerns about the impact of COVID-19 on themselves and their infant.

A brief communication by Ceulemans and colleagues [2] reported results on a perinatal mental health survey of 5866 women (2421 pregnant and 3445 breastfeeding). These feelings were true for both first-time mothers and mothers of multiple children. Self-reported major depressive symptoms (as assessed by the Edinburgh Depression Scale (EPDS); score ≥ 13) were reported in 812 breastfeeding women (23.6%). As for anxiety (as assessed by Generalized Anxiety Disorder 7-item Scale), 1620 (47.6%) reported minimal symptoms, 1306 (38.4%) mild, 317 (9.3%) moderate, and 161 (4.7%) severe. However, no more information is provided regarding its link with the breastfeeding experience or specific COVID-19 measures (e.g., isolation).

Only one study [44] evaluated the association between depressive symptoms as assessed by the EPDS and breastfeeding. Women who were exclusively breastfeeding scored significantly lower on the EPDS as compared to other feeding methods (e.g., formula). Regarding the current pandemic, mothers who were formula feeding had the highest EPDS scores. In addition, women who delivered during the pandemic and who scored above 12 on EPDS were more at risk of failing exclusively breastfeeding. Women in both groups who were exclusively breastfeeding presented significantly lower scores in the EPDS anhedonia and depression subscales. Finally, anxiety levels were similar in both groups.

## 4. Discussion

We reviewed the available literature regarding the impact of COVID-19 and related restrictions on breastfeeding and its impact on mental health outcomes for mothers. Although our main aim was focused on how changes in breastfeeding practices and expectations due to COVID-19 impact the mental health status and psychological functioning of mothers, the few results available did not allow for a thorough exploration of this association. Despite the increased interest in mental health outcomes during the COVID-19 pandemic (e.g., [37,46]), the association between changes in breastfeeding decisions and expectations and its impact on mental health outcomes is still only anecdotally observed. To overcome this gap in the literature we further explored related variables that would potentially offer critical information for the association between breastfeeding decisions/expectations and mental health, namely, safety concerns and women’s perception of support. Safety concerns include not only the sense of safeness regarding medical treatments (e.g., psychotropics) but also structural inequalities and disadvantages that impact women’s well-being (such as insecure employment, domestic violence, poverty, insecure immigrant status, etc.), all of which are well-known vulnerability factors for the development of emotional disorders in general and in the peripartum period in particular [47]. The perception of emotional, instrumental, and informational support includes the access and the availability of formal and informal parenting support (both for mothers and fathers); this is closely associated with perceived loneliness and is another protective factor for the development of perinatal depression and stress [48,49].

Overall, studies showed that the COVID-19 pandemic impacted expectations regarding breastfeeding, both positively (e.g., extending maternity leaves, offering mothers more time to enjoy motherhood) and negatively (e.g., restricting social and professional support), which in turn affected mothers’ mental health status and psychological functioning.

The negative influence of the COVID-19 pandemic on the breastfeeding experience was reported by a high number of women, even when not diagnosed with COVID-19 [2,36]. The main reasons reported for its negative impact are the increased childcare responsibilities at home and the lack of family, emotional, and professional support, which seem to have led to an increased experience of anxiety and stress related to breastfeeding. However, this interpretation requires caution as it seems not possible to disentangle whether anxiety and stress results solely from the perception of poor support or whether it is the result of the general impact of the ongoing pandemic. What is reported, however, is that the perceived poor support associated with isolation or inaccessibility to support groups reinforced the impact of COVID-19 on breastfeeding practices, since support groups are associated with a better adherence to breastfeeding [37] and consequently better mental health [46]. Additionally, the benefits of breastfeeding per se regarding mental health have been well documented, with studies showing that during breastfeeding hormones are secreted to act on the maternal central nervous system, leading to social responsiveness, maternal behavior, and proximity, as well as reducing physical and emotional stress responsivity [50,51,52]. As a result of the effects mediated by the oxytocin hormone, breastfeeding is also able to decrease the risk of postpartum depression and anxiety [53,54]. Therefore, the separation of the mother and infant and the consequent lack of breastfeeding can impact the perception and experience of “bonding”, leading to an increase in maternal stress during the postpartum period [55].

Nonetheless, although COVID-19 may have acted as an additional risk factor for mental health and vulnerability during the peripartum period, evidence also points to a few possible positive effects on breastfeeding perceptions [2,13,36]. In fact, for some mothers, this pandemic seems to have facilitated breastfeeding practices, namely, the increase in the duration of breastfeeding and the increased time at home to dedicate to the infant [2,36]. Interestingly, a high number of mothers across studies did not attribute a significant impact of COVID-19 on breastfeeding intentions and practices. This highlights that some women were able to cope effectively with difficulties arising from the COVID-19 pandemic. This may be associated with the mothers’ individual experience characteristics (e.g., support from family or health professionals). Exploring this influence may prove useful to the understanding of the efficacy of interventions and to better tailor it to mothers, particularly during the current crisis.

Overall, the current literature does not allow for the evaluation of the impact of COVID-19 on breastfeeding decisions by women and expectations and how these impact women’s psychological outcomes. However, the literature clearly suggests that COVID-19-negative mothers who had the opportunity to pursue their plans to breastfeed benefited from the pandemic, possibly increasing feelings of joy and accomplishment. However, COVID-19-positive mothers who were prevented from breastfeeding, but intended to, experienced feelings of loss (e.g., loss of control over their motherhood), frustration, or sadness. Additionally, for those mothers, the pandemic also impacted their trust in health services. Even though few results pointed to the negative psychological impact of COVID-19 in light of the breastfeeding experience, other factors may have potentially confounded the results. Separation itself has a negative impact on the overall mental health of the mother, and this relation is observed, for example, on mothers of premature infants outside pandemic periods who experience acute suffering [56].

The evidence suggests that health management services must consider the deleterious impact of separating mothers and infants when implementing policies and guidelines [25,55]. Adding to previous knowledge about the impact of separation of mothers and infants and considering the psychological impact of COVID-19 and related restrictions (including separation) on mothers’ mental health, psychological interventions should be incorporated in health-care protocols [41]. In fact, current guidelines formulated specifically to address the postpartum period during the COVID-19 pandemic strongly recommend the maintenance of breastfeeding practices [40,57,58]. However, contrary to these recommendations, parents frequently opted for formula and expressed breastmilk, which may be explained by the discrepant empirical evidence regarding vertical transmission of SARS-CoV-2 through breast milk, increasing uncertainty and anxiety [39].

Consequently, and especially for first-time mothers, support systems are needed to help with the breastfeeding and lactation process. Our review has highlighted the importance of postpartum breastfeeding care, as reported by mothers in general, and the need for its improvement. Specifically, health services should rethink how they provide this support during the pandemic, namely, through telehealth and in-person when possible. Communication of evidence-based information and access to technical assistance and support should be provided for all families [46,59].

Although we have identified a current gap in the scientific literature, local and international media seem to address this concern partly through news reports and social media posts of mothers’ own experiences. However, this media coverage may also have a negative impact on mental health as this information is, most of the time, built on subjective reports and not on evidence-based facts, increasing potential disinformation [60].

Several limitations to this literature review should be noted. Firstly, we included few electronic databases in our search. Secondly, the methodological quality of the reviewed studies was not assessed (therefore, publication bias was not controlled), and no peer-review process (external reviewer) was conducted in the screening of articles. Additionally, due to the few data available, no meta-analytic analysis was performed. As the majority of the included peer-reviewed studies were cross-sectional and observational, this does not allow for the inference of causality. In addition, the reviewed case-report provided important information, but its results may have been influenced by confounding factors, namely, the infant’s premature birth.

To face the limited number of available published studies and given the increased publications regarding COVID-19, the research team decided to include pre-prints. Although we are aware of the possible underlying bias and lack of quality checking, we opted to analyze the available information in these reports, selecting only pre-prints that focused on our target outcomes. The decision of including a source of grey literature (preprints) in this review is in accordance with the decision-making checklist proposed by Benzies and colleagues [61] to aid decisions about the inclusion/exclusion of grey literature. The included pre-print provided the collection of further information, allowing for a broader review and an increased chance of reviewing every possible source of relevant information.

Despite the above-mentioned limitations, this study was able to review the current state of the literature. Our findings can be used to inform good practices regarding breastfeeding decisions according to the COVID-19 status of mothers and accounting for other individual and social features that seem to (at least partially) determine its impact in women’s mental health. Hence, our study may also inform the development of guidelines for decision-making for parents and health providers regarding breastfeeding and mental health care. However, the observed gap in literature demands for further research to highlight and explore the objective association between the COVID-19 pandemic, breastfeeding expectations and practices, mental health status, and the psychological functioning of mothers.

## Figures and Tables

**Figure 1 behavsci-11-00034-f001:**
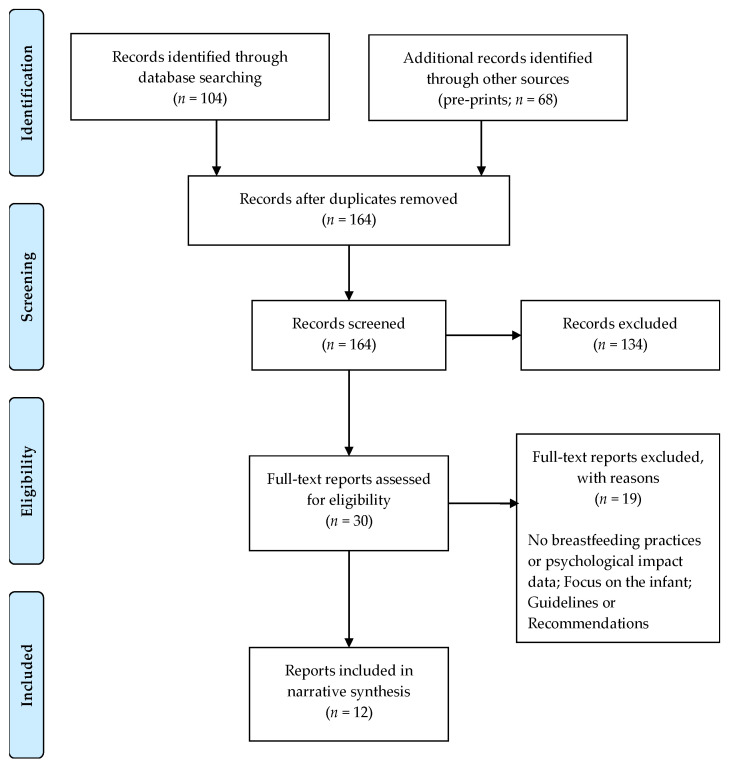
Flow diagram of the search and selection process.

**Table 1 behavsci-11-00034-t001:** Detailed information of the included studies.

**Study**	**Country**	**Study Design**	**Participants (N)**	**Mean Maternal Age (Years)**	**Infant’s Age**	**COVID-19 Diagnosis**	**Mental Health Outcomes**	**Main Findings**
[2]	Belgium	Cross-sectional	3445				Yes	Experience of anxiety and depression symptoms
[36]	Belgium	Cross-sectional	3823	32		*n =* 8	No	Overall, the COVID-19 pandemic only affected the duration and frequency of feeds; Perception of reduced support
[37]	United Kingdom	Cross-sectional	1049 (before lockdown)	31.7	M = 5.9 months	No	Yes	Reported loss of motherhood experiences; Perception of insufficient feeding support
			316 (during lockdown)	31.4	M = 1.27 months	No		
[13]	United Kingdom	Mixed methods	1290	30.92	M = 13.24 weeks	Unclear	No	Women reported both positive and negative breastfeeding experiences during the COVID-19 pandemic
[38] *	United States	Cross-sectional descriptive	258	30.7	N.A.	No	No	Women reported changing their breastfeeding plans due to the COVID-19 pandemic
[39]	Turkey	Multicenter cohort	125		30–39 weeks	*n* = 125	No	Lower rates of breastfeeding compared to formula and expressed breastmilk
[40] **	Australia	Cross-sectional	336			*n* = 25	Yes	Significant anxiety related to breastfeeding
[41]	India	Case report	1	26	28 weeks	*n =* 1	Yes	Feelings of guilt and decreased perception of efficacy
[42]	United States	Cross-sectional phenomenological qualitative	29	29.93	M = 3,86 months		Yes	Heightened stress; Negative influence of COVID-19 pandemic on support
**Study**	**Country**	**Study Design**	**Participants (N)**	**Mean Maternal Age (Years)**	**Infant’s Age**	**COVID-19 Diagnosis**	**Mental Health Outcomes**	**Relevant Results**
[43]	United States	Observational longitudinal cohort	85			*n =* 85	No	Separation of mother–infant impacted negatively on breastfeeding rates
[44]	Italy	Non-concurrent case control	152 (delivered during lockdown)	33.47			No	Lower depression symptoms in mothers who breastfed
			147 (delivered in 2019)	33.18			Yes	
[45]	United States	Case series	3	29	Up to 67 days	No	No	Mothers experienced disruptions in support
				31	Up to 58 days	No		
				33	Up to 40 days	No		

N.A. = Not Applicable. * Although this study included pregnant women itwas included in this review because it reported changes to breastfeeding plans. ** Pre-print.

## Data Availability

No new data were created or analyzed in this study. Data sharing is not applicable to this article.

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
