# Peer review of "Breastfeeding during COVID-19: A Narrative Review of the Psychological Impact on Mothers"

_behavsci, 2021, doi:10.3390/bs11030034_

Round 1

Reviewer 1 Report

Thank you for giving me the opportunity to review this timely and interesting paper. Overall, the paper properly addresses a significant topic related to the pandemic outcomes and their impact upon families and, most importantly, the relationship between mothers and their children. However, there are some aspects that need improvement, mostly in the Introduction part:

1) The negative impact of these policies (e.g., separation) on maternal mental health outcomes is anticipated to be higher, - lines 40-42 / This sentence is unclear.

2) Besides culture and sociodemographic conditions, the decision on breastfeeding is also influenced  by other psychosocial factors such as precarious work, maternity leave duration, and willingness of the father to actively participate in the newborn care tasks (lines 47-49) - how, exactly, do these variables impact the breastfeeding choice? Please detail.

3) Additionally, breastfeeding is frequently a difficult experience for mothers (line 50) - please detail using proper references.

4) This study was conducted in 107 webpages/online forums and found three defining themes that contribute to the decision of breastfeeding or  adopting formula milk. - lines 55 to 57  -this needs further clarification (and, probably, rephrasing).

5) During global infection outbreaks such as the COVID-19 pandemic, however, this sense of “free parenthood” is frequently put-on hold - lines 61 to 63 - what does free parenthood actually mean? Please detail using proper references.

6) Also, the extensive media coverage can  induce fears in new parents and clinicians related to both formula and breastfeeding practices, changing perspectives and assumptions during a global health crisis (lines 70-72) - how, exactly, does the media coverage during COVID-19 impact breastfeeding/formula practices? Please detail.

7) Lines 86-88: has identified a significant increase in mental health symptomatology, particularly in depressive (Edinburgh Postnatal Depression  Scale; EPDS [19]) and anxiety (State-Trait Anxiety Inventory; STAI-State [20]) outcomes. - I would not recommend outlining the specific instruments the authors have used unless they are highly related to the present study or the authors want to point out certain differences or similarities related to the current findings/research procedure.

Thank you.

Author Response

Thank you so much for your comments, please see the detailed author response by attachment.

Reviewer 2 Report

Thank you for this interesting and useful manuscript.

Line 132: You mention a rapid systematic search - can you clarify any particular deviations from accepted systematic review protocols?

Line 148-149: Perhaps summarise the types of data you specifically sought from authors.

Line 225: Perhaps specify here that COVID-19 was not reported as a significant factor in changing plans to breastfeed. The mention of the mother with depressive symptoms doesn't seem relevant here, as this doesn't appear to be related to COVID-19.

Section 3.1.2: Are these the only two studies that discussed changing the feeding plan? If any other studies explored this but reported that COVID-19 was not responsible for changing the feeding plan, please note this here.

Line 263: You say "for every moment" - do you mean "for every mother"?

Line 265-266: You say separation and difficulty latching due to COVID-19 - it's unlikely COVID-19 affected latch significantly, so perhaps re-phrase to "separation due to COVID-19 and difficulty latching".

Line 305: Many of the issues mentioned in this paragraph are found in any study on the experiences of women when breastfeeding. Can you elaborate more on whether the study was able to demonstrate that COVID-19 was an independent predictor of the issues mentioned?

Line 350: End of the sentence is cut off.

Line 367: How might these figures compare to non-COVID times?

Line 379: You mention stress caused by the pandemic - did this study have causal data, or was this stress reported by mothers?

Line 448: It may also be useful to acknowledge the sometimes large number of women who reported no significant impact of COVID-19 on their breastfeeding plans and practices. This is important in acknowledging the effectiveness of some existing supports, and also highlights that some women did manage effectively throughout the pandemic, and closer examination of their experiences may be of benefit.

Author Response

(The authors gave the same response as above.)
